# The Evolving Landscape of Cytoreductive Nephrectomy in Metastatic Renal Cell Carcinoma

**DOI:** 10.3390/cancers15153855

**Published:** 2023-07-29

**Authors:** Hana Studentova, Martina Spisarova, Andrea Kopova, Anezka Zemankova, Bohuslav Melichar, Vladimir Student

**Affiliations:** 1Department of Oncology, Faculty of Medicine and Dentistry, University Hospital Olomouc, Palacky University, 771 47 Olomouc, Czech Republic; hana.studentova@fnol.cz (H.S.); martina.spisarova@fnol.cz (M.S.); andrea.ondruskova1@gamil.com (A.K.); anezka.zemankova@fnol.cz (A.Z.); bohuslav.melichar@fnol.cz (B.M.); 2Department of Urology, Faculty of Medicine and Dentistry, University Hospital Olomouc, Palacky University, 771 47 Olomouc, Czech Republic

**Keywords:** renal carcinoma, cytoreductive nephrectomy, deferred nephrectomy, immunotherapy, tyrosine kinase inhibitors

## Abstract

**Simple Summary:**

For a significant period of time, the removal of the primary tumor termed cytoreductive nephrectomy has been considered the standard of care in patients with metastatic renal cell carcinoma. The situation is complicated because of a very quickly changing landscape of systemic therapy in metastatic renal cell carcinoma. After the turn of the century, cytokines were substituted by multiple tyrosine kinase inhibitors that dominated the therapy of renal carcinoma for more than a decade. With the expansion of immune-based systemic therapy, the importance of cytoreductive nephrectomy has been widely discussed and often disputed. Due to the absence of prospective data regarding the role of cytoreductive nephrectomy in the immunotherapy era, we can at this moment rely only on retrospective studies with relatively small numbers of patients. Nevertheless, with an individualized approach, we should attempt to identify in the clinical practice patients with favorable prognostic patterns who might benefit from the combination of surgery with systemic treatment.

**Abstract:**

The role of cytoreductive nephrectomy in metastatic renal cell carcinoma (RCC) has been studied intensively over the past few decades. Interestingly, the opinion with regard to the importance of this procedure has switched from a recommendation as a standard of care to an almost complete refutation. However, no definitive agreement on cytoreductive nephrectomy, including the pros and cons of the procedure, has been reached, and the topic remains highly controversial. With the advent of immune checkpoint inhibitors, we have experienced a paradigm shift, with immunotherapy playing a crucial role in the treatment algorithm. Nevertheless, obtaining results from prospective clinical trials on the role of cytoreductive nephrectomy requires time, and once some data have been gathered, the standards of systemic therapy may be different, and we stand again at the beginning. This review summarizes current knowledge on the topic in the light of newly evolving treatment strategies. The crucial point is to recognize who could be an appropriate candidate for immediate cytoreductive surgery that may facilitate the effect of systemic therapy through tumor debulking, or who might benefit from deferred cytoreduction in the setting of an objective response of the tumor. The role of prognostic factors in management decisions as well as the technical details associated with performing the procedure from a urological perspective are discussed. Ongoing clinical trials that may bring new evidence for transforming therapeutic paradigms are listed.

## 1. Introduction

Renal cell carcinoma (RCC) is one of the most common urological malignancies [1,2]. The widespread use of ultrasound and computed tomography (CT) has led to an increasing proportion of RCCs detected and treated at an early stage [3]. However, nearly a fifth of patients present with synchronous metastases, and the issue of radical nephrectomy in these patients remains quite controversial [4]. RCC is characterized by a high level of vascularity and high immunogenicity [5,6]. These facts, along with a lack of sensitivity to cytotoxic chemotherapy, represent the rationale as to why the treatment of RCC is not based on standard cytotoxic agents in contrast to most other solid tumors. In the era of cytokines, cytoreductive nephrectomy (CN) was considered to be the gold standard of care [7,8]. A combined meta-analysis of these trials indicated a statistically significant median OS benefit of 5.8 months in patients treated with interferon alpha (IFN-α) plus CN compared to IFN-α alone [9]. With the advent of vascular endothelial growth factor (VEGF)-targeted therapy, survival has been prolonged considerably and the role of nephrectomy has been questioned. With regard to the registration trials of these novel agents, a significant proportion of patients enrolled had undergone CN before entering the study [10,11,12,13,14,15]. In addition, a number of retrospective trials noted a survival benefit of CN plus targeted therapy versus targeted therapy alone [16], and a systematic review of 10 trials reported improved OS in patients with CN [17]. The results of two large and widely cited trials, CARMENA (Cancer du Rein Métastatique Néphrectomie et Antiangiogéniques) and SURTIME, contributed to a great extent to the modification of the view on upfront CN. CARMENA was a prospective trial demonstrating the non-inferiority of sunitinib compared to sunitinib plus CN. The median OS was 18.4 months in patients treated by sunitinib alone compared to 13.9 months in patients who underwent upfront CN plus sunitinib [18]. Nevertheless, for several reasons, the results of the CARMENA trial must be interpreted very cautiously. Most importantly, the trial lacks the risk-adapted approach described by Arora et al. [19] due to the fact that low- burden disease patients were offered surgery outside the trial. The updated data from the CARMENA trial support the notion that CN should not be the standard of care for all patients with metastatic RCC. It is suggested that some patients from the intermediate-risk group may benefit from upfront CN. Benefits from this approach could be dependent on a number of International Metastatic RCC Database Consortium (IMDC) risk factors (patients with only one risk factor seem to benefit from CN plus sunitinib) or localization of metastases (patients with lung metastases only seems to be good candidates for CN) [20]. The cohort of patients from the sunitinib-only group who underwent deferred CN (18%) had a significantly better outcome with a median OS of 48.5 months vs. 15.7 months (HR: 0.34; 95% CI 0.22–0.54) in comparison with those who remained on sunitinib only [20]. Moreover, Kutikov et al. published an analysis reporting that 30% of CN patients were unable to receive systemic therapy after CN due to disease progression or perioperative mortality in approximately half of the cases [21].

Another prospective phase III trial, SURTIME, enrolled 99 instead of the initially planned 458 patients. Patients were randomized to immediate CN with a subsequent four cycles of sunitinib or to three cycles of sunitinib followed by CN and two adjuvant cycles of sunitinib. Patients in the deferred CN arm who progressed after 3 cycles of sunitinib were not indicated to CN. Patients who received sunitinib prior to CN had longer OS versus patients who had immediate CN (median 32.4 months vs. 15 months; HR, 0.57; 95% CI, 0.34–0.95) There was no difference between the two study groups, and this trial did not meet its primary endpoint (13). Although it may seem that patients with a good response to systemic treatment could be candidates for delayed CN, other prospective trials are needed to evaluate the extent of potential benefits.

Multiple tyrosine kinase inhibitors (MTKI) and immune checkpoint inhibitors (ICI) have completely transformed the way of metastatic RCC management [20]. New promising combinations of MTKIs and ICIs have become a new standard of first-line treatment [18,22]. In the same year in which results from CARMENA trial were published, high activity of the combination of ipilimumab with nivolumab in the intermediate- and poor-risk metastatic RCC patients have been reported [23,24] In the light of the recent data from trials combining ICIs (CheckMate 214, combining ipilimumab with nivolumab) or ICI plus MTKI (CheckMate 9ER, combining nivolumab with cabozantinib; CLEAR, combining pembrolizumab with lenvatinib; Keynote 426, combining pembrolizumab with axitinib, or Javelin 101, combining avelumab with axitinib), the impact of the results of the CARMENA and SURTIME trials is becoming less clearly defined [25,26,27,28,29,30,31,32] All the listed combinations have shown a significant benefit in all evaluated parameters over sunitinib monotherapy. Meanwhile, survival data reported from the aforementioned clinical trials did not consistently stratify patients with regard to prior CN. Potential survival benefits of systemic therapy are also accompanied by high cost and chronic toxicity. Rarely do these therapies lead to complete responses that result in a permanent cure. Looking at the data in more detail, the progression-free survival (PFS) benefit was observed in patients undergoing prior nephrectomy in the ipilimumab plus nivolumab combination and nivolumab plus cabozantinib, with no benefit noticed in patients treated with pembrolizumab plus axitinib [26,30,33]. With regard to OS, a survival benefit associated with prior nephrectomy status was confirmed only in patients treated with nivolumab plus cabozantinib, but not in nivolumab plus ipilimumab or pembrolizumab plus axitinib-treated patients. However, the analyses did not distinguish between patients with nephrectomy for early RCC in the past versus CN in patients with synchronous metastases. Moreover, the trials were underpowered for the analysis of subgroups, including the presence or absence of nephrectomy. Hence, despite exciting results from the trials of new combination regimens, these do not answer the question regarding the role of cytoreductive surgery in metastatic RCC. Yet, CN is still being considered as we have not had reliable predictive factors for immunotherapy treatment combinations [32,34,35,36,37].

## 2. Current Status of Cytoreductive Nephrectomy in the Immunotherapy Era

In the context of the evolution of ICI-based combinations over the past decade, the role of CN has been substantially transformed. Indeed, the immunotherapy era evidence regarding the combination of CN with systemic treatment, patient selection, and the timing of surgery remain insufficient for the moment. Recommendations from leading oncology and urology societies are based on the data coming from the targeted therapy era. Currently, only limited evidence is available to determine who could be considered an appropriate candidate for CN in combination with ICI-based regimens. Some ongoing prospective trials evaluate the outcomes of ICI containing therapy in combination with or without CN [38], out of which PROBE (NCT04510597), NORDICSUN (NCT03977571), or Cyto-KIK (NCT04322955) may answer some questions with regard to the choice between immediate versus deferred CN in the immunotherapy era. However, the results from these trials are not available yet [39].

The fundamental rationale for CN resides in the reduction of tumor burden aiming at decreasing the number of tumor cells potentially resistant to systemic therapy. Other potential mechanisms include an indirect effect on tumor microenvironment, metabolic acidosis caused by reduction of functional nephrons, or a decrease of antiangiogenic factors following nephrectomy. What exactly is behind the potential life-prolonging effect of CN yet remains unclear [16,40,41]. In the light of the often contradictory data, defining the role of CN in the current era remains a clinical challenge [42,43]. The National Cancer Database has been used in several population-based cohorts to assess the impact on CN in the context of immunotherapy. Singla et al. reported results from a retrospective study comparing outcomes in patients treated with CN plus ICIs versus ICIs alone. Out of 391 clear cell RCC (ccRCC) patients (including 5.6% patients with sarcomatoid histology), survival benefit has been reported in 221 who underwent CN plus ICIs over patients treated with ICIs alone (HR 0.23, *p* < 0.001). The preoperative administration setting of ICIs improved the outcome not only in terms of downstaging the tumor stage and grade but also by increasing the number of patients achieving pathologic complete response (pCR) in the primary tumor (10% patients). It should be noted that patients undergoing upfront CN had more favorable tumor characteristics. Nevertheless, this is the first report that shows survival benefits in metastatic RCC patients treated with ICIs along with CN and accentuates the role of CN in the modern therapy era [44]. On multivariable analysis, no predictors of favorable outcomes regarding CN timing were identified in this cohort. Bakouny et al. reported data from a retrospective multicenter analysis of de novo 4639 metastatic RCC patients (including non-clear cell and sarcomatoid histology) treated with systemic therapy including ICIs (437) or targeted therapy (4202). A meaningful proportion of patients in the ICI arm (54%) and targeted therapy arm (55%) received CN. Significantly better OS was identified with upfront CN in both ICI and targeted therapy groups. In the ICI group, the median OS was 54 months (95% CI, 34–not reached) in the CN group versus 22 months (95% CI, 17–25) in the group without CN, and 25 months (95% CI, 23–26) versus 13 months (95% CI, 12–14) in the targeted therapy group. There was no difference in the extent of survival benefits connected with CN between the ICI and targeted therapy groups. The study confirms the OS benefit of upfront CN in selected patients and has an important place in the management of the disease [45]. Another multicenter retrospective review of data from the Seattle Cancer Care Alliance and The Ohio State University was published in 2022. Outcomes of patients diagnosed with metastatic RCC between 2000 and 2020 and treated by ICIs in any time of their treatment course were evaluated. The study included patients with both upfront (202) and deferred (30) CN. Comparison between the two cohorts yielded a substantial median OS difference of 56.3 months vs. 19.1 months in CN plus ICI vs. the ICI-alone group. There were no significant differences in OS among primary or deferred CN. Longer OS was observed in patients treated by CN with ICIs in any line of therapy [39]. Interestingly, Pieretti et al. reported better survival outcomes in metastatic RCC patients with an intermediate-risk score while achieving metastatic tumor shrinkage of at least 10% after preoperative therapy (TKI, ICIs, or both) followed by CN [46]. Nevertheless, there is no clear evidence of CN indication and timing, and the role of CN remains a matter of debate. An individual approach including optimal timing should be discussed in a multidisciplinary team.

Data from publications investigating the role of CN in patients treated exclusively with ICIs or ICI-based combinations are summarized in Table 1.

## 3. Cytoreductive Nephrectomy in Non-Clear Cell Subtypes

Metastatic non-clear cell carcinomas (nccRCC) portend generally worse prognosis than ccRCC [53]. The role of CN in nccRCC is rather uncertain due to the rapid evolution of systemic therapy and sparcity or even the lack of prospective or even retrospective data [54,55]. The optimal management of metastatic nccRCC remains largely questionable in the absence of prospective randomized trials. Several large retrospective observational studies showed improved outcomes in patients treated with both CN and systemic treatment regardless of histology [56,57,58], supporting CN in patients with metastatic nccRCC [59].

The systemic therapy of papillary RCC has gone through some progress tracking the development in ccRCC, however, the optimal treatment strategy remains mostly undefined. Discussing the role of CN, Riveros et al. reported improved OS in patients with metastatic papillary RCC treated with ICI-based therapy or targeted therapy in combination with CN [60]. While the impact of the histological subtype seems to be critical in the systemic treatment selection, determining the role of CN based on the differences arising from molecular background may not be that plausible [60].

However, data on CN in patients with non-clear cell histology are still sparse and exclusively retrospective. In summary, we have no data to draw any recommendations with a regard to CN in metastatic nccRCC.

The results of retrospective data need rapid extrapolation as it will take some time to obtain relevant data from prospective trials. Nonetheless, the role of accurate patient selection is unambiguously crucial. Defining the role of CN in the era of ICIs warrants prospective validation in clinical trials.

## 4. Cytoreductive Nephrectomy in Sarcomatoid RCC

Sarcomatoid RCC represents a relatively uncommon entity associated with poor prognosis with median OS less than one year in a vast majority of cases [61,62]. Sarcomatoid dedifferentiation can be detected in any histological variant in contrast with rhabdoid which occurs exclusively in ccRCC [52]. The importance of CN has been often doubted in this patient population because of unfavorable outcomes [55]. Nevertheless, in patients with sarcomatoid RCC undergoing CN, an improved outcome was also reported compared to patients with no surgery [54]. On the other hand, poor prognosis and hardly any benefit from CN in patients with sarcomatoid dedifferentiation has been published by Adashek et al. [63]. The authors point out that in cases of unfavorable histology such as sarcomatoid on pretreatment biopsy, systemic therapy should be initiated as the only meaningful strategy with potential benefit to the patient [64]. Hahn et al. published data showing no statistically significant benefit of CN in terms of prolongation of ICI therapy or OS for CN in patients with sarcomatoid or rhabdoid dedifferentiation [52]. Nevertheless, the authors claim that there might be a subset of patients who derive substantial benefit from CN. That is in concordance with a recently published small series of patients with sarcomatoid RCC and poor prognostic features who derived benefit from immediate CN resulting in durable treatment response [65].

In the case of sarcomatoid or rhabdoid RCC, we should seriously re-evaluate the role of CN in the initial management of patients with this rare presentation who derive great benefit from ICIs. In particular, in cases of a large primary tumor which is often encountered upon tumor occurrence, it could be speculated whether debulking of the large primary tumor mass could affect the response to subsequent ICIs [65]. Not forgetting to mention, the European Society for Medical Oncology (ESMO) guidelines recommend CN in cases of large primary tumors [66].

## 5. Surgical Aspects of Cytoreductive Nephrectomy

CN is associated with significant perioperative morbidity and perioperative (90-day) mortality reported in the range of 0–10.4% [67,68]. The probability of perioperative mortality is significantly higher in older patients (≥71 years), patients with multiple comorbidities (Charlson Comorbidity index, CCI ≥ 2), and frail patients [69]. The reported overall complication rate ranges from 11.5 to 54.5% [70,71]. Severe complications (Clavien ≥ 3) occur in 3–36.4% of cases [71,72]. The most frequently reported complications include bleeding requiring blood transfusion (30.8%), infectious complications (9.8%), venous thromboembolism (2.7%), and cardiac complications (1.7%) [73]. Gershman et al. identified liver metastases, the need for intraoperative blood transfusions, and the pN1 stage as factors associated with higher perioperative morbidity in 294 CN patients [74]. Tanaka et al. identified the clinical T stage as a predictor of perioperative complications [75]. CN is also associated with an increased severe complication rate (Clavien ≥ 3) compared to nephrectomy in localized disease (7.3% vs. 3.2%, *p* < 0.0001) [73].

In order to optimize selection, several studies have made an effort to identify optimal candidates for immediate or deferred CN. New prognostic models aiming at better patient selection for CN have been described. In 2010, a retrospective analysis of the M.D. Anderson Cancer Center (MDACC) institutional RCC database was conducted which included 566 metastatic RCC patients who underwent CN and 110 patients treated with systemic therapy alone [76]. The analysis revealed seven factors including clinical criteria such as symptoms from metastasis (e.g., bone pain), T3/T4 primary tumor, the presence of liver metastasis, and retroperitoneal or supradiaphragmatic adenopathy, all present at the time of CN, to be associated with an increased risk of death. With a regard to laboratory parameters, these included serum albumin concentration below the lower limit of normal and the level of serum lactate dehydrogenase (LDH) above the upper limit of normal. According to this study, patients with ≥4 factors were unsuitable candidates for CN [77]. This group from MDACC recently published an update of this model calculated on 608 patients and operated between 2005 and 2017. This model can be used to calculate preoperative risk factors associated with an increased risk of death in metastatic RCC patients undergoing CN. The authors identified nine altered preoperative factors associated with higher overall mortality including the presence of symptoms, retroperitoneal lymphadenopathy, supradiaphragmatic lymphadenopathy, bone metastases, clinically T4 primary tumor, anemia, hypoalbuminemia, LDH elevation, and increased neutrophil to lymphocyte ratio (NLR). The authors divided patients into three groups based on these factors: low- (<2 factors), moderate- (2–3 factors), and high-risk group (>3 factors) which could help to select patients less likely to derive benefit from surgical approach. OS in these groups was 58.9, 30.6, and 19.2 months, respectively. The authors further observed an association of groups with unfavorable final pathologies such as sarcomatoid and rhabdoid dedifferentiation, lymphovascular invasion, and the presence of necrosis in the tumor specimen with the risk of death. On the other hand, these prognostic factors can also herald the presence of a tumor with unfavorable biology. In addition, these factors have been associated with adverse perioperative outcomes such as blood loss, complication rates, and rehospitalization [78]. Another predictive model was developed by the Registry for Metastatic RCC (REMARCC) group based on a retrospective analysis of 519 patients operated on between 2005 and 2019. This model identified obesity as a predictor of lower overall mortality, HR 0.56, *p* = 0.007. Conversely, bone (HR 1.49, *p* = 0.01), liver (HR 1.71, *p* = 0.002), and lung (HR 1.6, *p* < 0.001) metastases were associated with increased mortality. Another factor associated with higher mortality was a worse overall condition, performance status <80% (HR 1.5, *p* = 0.026) [79]. So far, none of the evaluated factors or models is universally used, although the selection of patients who might benefit from CN is very important. Kutikov et al., for example, found that up to 30% of patients could not undergo systemic treatment after previous CN due to disease progression or death [21]. Furthermore, Silagy et al. reported an analysis evaluating the change in IMDC criteria before and six weeks after CN. Individual risk factors changed in both directions and remained unchanged in only 42.6% of patients. An improvement in OS (hazard ratio = HR 0.64, *p* = 0.007) was found in the group with a reduction of risk factors (28.2% of CN patients). On the contrary, in the group with an increase (25.6% CN patients), OS worsened (HR 1.57, *p* = 0.007). Changes occurred mainly in laboratory parameters but also in the performance status. In descending order, the most frequent changes noted concerned hemoglobin, neutrophils, platelets, KPS, and calcium. However, the normalization of parameters occurred most commonly in calcium and least commonly in hemoglobin levels [80]. At the same time, it is unclear whether the surgery changes these factors or whether CN affects metastatic RCC itself. Prognostic models of CN feasibility that assess several unfavorable preoperative parameters. are summarized in Table 2.

The benefit of removing a small primary tumor could be questioned. The size of the primary tumor in the context of its removal was addressed by Tappero et al. [81]. The authors published favorable results within the The Surveillance, Epidemiology, and End Results (SEER) database analysis demonstrating a strong association between CN and OS in metastatic RCC patients with a primary tumor size of ≤4 cm regardless of tumor histology or systemic therapy exposure. While these data are in agreement with previously published work on the benefits of CN, this is the first study to report benefits in patients with small primary tumors. However, the data do not discuss the timing of CN nor the potential role of ICIs in this patient population.

Partial nephrectomy (PN) as a surgical procedure is rarely used in metastatic RCC, but may be useful, particularly in the case of bilateral tumors or in patients with a solitary kidney. Data from the SEER database and the National Cancer Database (NCDB) show a PN rate of 4.2% and 3.8%, respectively [82,83]. PN did not show inferior oncological outcomes compared to radical nephrectomy (RN) in four studies [82,84,85,86] and was, on the contrary, associated with better outcomes in three published sets [9,82,87]. The rate of early (30 days) complications in PN in solitary kidneys was higher compared to RN (33% vs. 10%, *p* = 0.009) [86]. Conversely, Mazzone et al., evaluating 217 PNs and 5171 RNs, found no significant difference in complication rates [82].

Minimally invasive CN is a commonly used method. For example, in the CARMENA study, 40% of patients underwent laparoscopic CN (LCN) [18]. In the work of Zlatev et al., which evaluated 24,145 CNs performed in the United States of America in the years 2003–2015, it showed a trend in reducing the rate of open CN (OCN) from 76.7% to 66.4%, LCN from 22.3% to 11.4% and, conversely, an increase in robotically assisted CN (RaCN) from 0.6% to 22.1% [88]. LCN, compared to OCN, has demonstrated better perioperative outcomes (lower blood loss and shorter recovery time) and non-inferior oncological outcomes in several smaller studies [89,90,91,92]. LCN was also not associated with greater morbidity in delayed CN [93]. Several studies have shown a lower rate of perioperative complications in LCN vs. OCN [68,70,73] and RaCN vs. OCN [94]. While in an analysis of the British Association of Urologic Surgeons (BAUS) database, the conversion rate from LRP to ORP in CN was 14%, in a study by Bragayrac et al., the conversion rate was only 3.3% [95]. The oncological results of minimally invasive CN were evaluated by Zhao et al. when comparing 48 LCN and 48 OCN. They found a longer OS in LCN (23.9 vs. 10.8 months, *p* < 0.01) [96].

Another factor studied is the presence of a tumor thrombus. Abel et al. demonstrated that a tumor thrombus extending into the inferior vena cava above the diaphragm is associated with worse OS than a thrombus in the renal vein alone (median 9.2 vs. 21.7 months, *p* = 0.0165) [97]. On the other hand, the extent of the tumor thrombus was not a predictor of OS in the study by Miyake et al. [98]. Kwon et al. published the results of 45 patients treated for metastatic RCC with tumor thrombus treated with CN plus systemic therapy (*n* = 28) compared to systemic treatment alone (*n* = 17). Median OS was 17.3 and 19.7 months in these groups (*p* = 0.0353), respectively. Thus, CN did not improve OS in metastatic RCC with tumor thrombus [99]. Conversely, Qi et al., in a large cohort of similar size, reported a median OS of 22 months in patients treated with CN + systemic therapy but only 12 months in patients treated with systemic therapy alone and six months in patients who underwent CN alone (*p* < 0.001) [100].

The presence of lymphadenopathy in metastatic RCC is associated with aggressive tumor biology and is a prognostic factor for worse PFS and cancer-specific survival [101,102]. Kroeger et al. confirmed that patients with nodal metastases have worse cancer-specific survival (*p* < 0.001) and OS (*p* < 0.001) compared to patients without nodal metastases. However, only subdiaphragmatic LNMs were a predictor of shorter OS according to this study (*p* < 0.001) [101]. Nevertheless, lymphadenectomy (LND), in the study of Gershman et al., did not lead to better oncological outcomes, even in the group with extended LND (≥13 nodes removed) [102]. Similarly, a systematic review did not reveal the benefits of performing LND for CN in patients with metastatic RCC [103]. The only benefit could be the value as a prognostic tool, as demonstrated by two other studies [104,105].

In the era of MTKI therapy, wound healing complications were of concern, and local wound healing complications after CNs were reported [68]. ICI-based therapy may not affect the process of wound healing, but the surgical procedure itself could be a challenge due to fibrotic changes induced by the tumor response [43,106,107,108]. ICI-based therapy can result in desmoplastic reaction increasing perinephric adhesions, and inflammation and, thus, surgical complexity [109]. Graafland et al. demonstrated the safety of performing CN in 21 patients after previous ICI treatment. The authors also did not observe a relationship between tumor size reduction and the rate of subsequent fibrosis [110]. Pignot et al., in the study of 11 patients operated on in eight French centers, described difficult dissection in patients treated with ICIs (in 82% of cases) due to inflammatory tissue reaction and adhesions [107]. In contrast, Singla et al. did not describe any problems in 11 cases of CN [111].
cancers-15-03855-t002_Table 2Table 2Prognostic models of CN feasibility that assess several unfavorable preoperative parameters.Prognostic ModelsYearFactorsNumber of Factors Suitable for CNRef.Culp et al. 2010Albumin < LLNcT3 or cT4 cN+ Clinical symptomsLiver mets<4 factors[76]Ohno et al. 2014NLR > 4 ECOG > 1
[75]You et al. 2014Hemoglobin < LLNNeutrophils > ULNKarnofsky performace status scale < 80cN2 (metastases ≥ 1 regional lymph node according to AJCC 2002) <2 factors[112]Fukuda et al. 2018Glasgow prognostic score:0: CRP 10 mg/L and albumin 35 g/L 1: CRP > 10 mg/L or albumin > 35 g/L 2: CRP > 10 mg/L and albumin < 35 g/LGlasgow < 2[113]McIntosh et al. 2020Albumin < LLNLDH > ULNHemoglobin < LLNNLR > 4cT4 Retroperitoneal lymfadenopathy Supradiaphragmatic lymfadenopathyClinical symptoms Bone mets<4 factors[78]Marchioni et al. 2021REMARCC:Normal weight Bone metsLiver metsLung metsNumber of mets ≤ 3 Karnofsky performace status scale < 80Good prognosis (0 factors) Intermediate prognosis (1–2 factors) Poor prognosis (>2 factors) [79]CN, cytoreductive nephrectomy; Ref., reference; NLR, neutrophil to lymhocyte ratio; LLR, lower limit of normal; ULN, upper limit of normal; CRP, C reactive protein; LDH, lactate dehydrogenase.


## 6. Recommendations from International Guidelines

Cytoreductive nephrectomy should be considered in patients with a primary tumor suitable for surgery and resectable oligometastatic disease. For the majority of metastatic patients, systemic treatment remains the principal therapeutic modality. International guidelines deal with CN in different ways. The European Association of Urology (EAU) guidelines mention two phase III studies, CARMENA and SURTIME trials, with regard to the sequence of CN and sunitinib. The guidelines currently highlight the paradigm shift associated with the era of ICIs and the ICI combinations with TKIs. The EAU guidelines point out that there is still a lack of high-level data-based evidence recommendations for CN in combination with ICI-based therapies and mention a systematic review evaluating effects of CN that demonstrated an OS advantage of CN in patients who do not need immediate systemic treatment [17]. These results were supported by a registry study showing that selected patients with primary CN had a significant OS benefit [114]. CN is not recommended in patients with poor prognostic features according to MSKCC, and immediate CN should not be performed in intermediate-risk patients requiring systemic therapy based on several retrospective studies [44,115,116], but also the prospective CARMENA trial [18]. On the other hand, the immediate CN should be offered to patients with a good performance status who do not require systemic therapy or patients with oligometastatic disease when radical local treatment of the metastases can be achieved. Meanwhile, a delayed CN approach should be discussed with patients who may derive clinical benefit from upfront systemic therapy as suggested by the results of the SURTIME trial [23,117].

According to the National Comprehensive Cancer Network (NCCN) guidelines, patients most likely to benefit from immediate CN are patients with oligometastatic disease which is manageable by local methods (lung, bone, or brain), good prognostic features, and good performance status. Patients with metastatic disease and symptomatic primary tumors (hematuria or other symptoms) should only be offered palliative nephrectomy if they are surgical candidates [118].

The ESMO guidelines underscore the role of a multidisciplinary team. The ESMO guidance does not recommend the immediate CN in Memorial Sloan Kettering Cancer Center (MSKCC) intermediate- and poor-risk patients with asymptomatic primary tumors. A deferred CN remains an option for patients with local symptoms and near complete responses to systemic treatment [66,117].

Current guidelines do not reflect the present situation with a regard to available combination therapies. The recommendations do not provide any general criteria for patient selection to an immediate or a deferred CN either as data regarding patients treated with modern systemic treatment still remain deficient.

## 7. Discussion and Future Perspectives

Defining an individualized strategy and patient selection are undoubtedly a crucial step in the process. Metastatic RCC is an extensively heterogenous disease with variable biological behavior, outcome, and unpredictable response to therapy. CN may be or may be not fundamental in the therapeutic strategy depending on multiple scenarios which are determined principally by specific disease characteristics such as the extent of the disease (low/high volume), size of the primary tumor, IMDC (or any other risk stratification factors), number and localization of metastatic sites, and last but not the least, patient performance status.

In addition, if we believe that CN should occur at some point, an important issue is the timing—whether the patient may benefit from the immediate CN or whether to go for a deferred surgery. Both options have advantages and disadvantages. The often mentioned potential risk of immediate CN is the postponing of effective systemic therapy because of complications of surgical procedures in this patient population. CN after initial systemic therapy may help to identify the patient subset most likely to benefit and potentially allow for the eradication of immune-resistant clones within the primary tumor. Unsurprisingly, early phase studies showed promising efficacy and safety data of a neoadjuvant approach as well as changes to immune infiltrates within the tumor providing a rationale for upfront systemic therapy. Having the tumor in situ or higher disease burden magnifies the immune response mediated by ICIs [119]. Benefits of neoadjuvant ICI administration have been recently demonstrated in patients with melanoma [120]. The assessment of tumor response to systemic therapy may be helpful in the selection of the appropriate candidates for the procedure as well as the timing of CN. Not surprisingly, Navani et al. published an analysis of 1085 patients with synchronous metastatic RCC treated by ICI-based therapy. In a multivariable analysis, favorable risk group, CN, and especially deferred CN were associated with an increased likelihood of ORR [121]. It is therefore not surprising that, currently, ongoing prospective trials did not include an arm with upfront CN.

There are currently two phase III trials ongoing in primary metastatic RCC patients, with the aim to assess the role of deferred nephrectomy with primary ICI therapy. In NORDICSUN (NCT03977571), a phase III multicenter randomized trial synchronous metastatic ccRCC or non-ccRCC patients without any prior systemic treatment are treated for 3 months or up to 4 cycles of nivolumab plus ipilimumab or MTKI/ICI combination. Subsequently, in patients with ≤3 IMDC risk factors, the multidisciplinary team decides whether the tumor is considered resectable. The patients deemed resectable are to be randomized between CN or continue with systemic therapy for 3 months more after which a second evaluation takes place. Patients who are still unresectable continue with nivolumab or MTKI/ICI maintenance. The primary endpoint of this trial is OS, principal secondary endpoints include progression-free survival (PFS), objective response rate (ORR), and time to subsequent systemic therapy [38]. The results are expected in 2026.

Another prospective phase III trial again evaluates the outcomes of CN in metastatic RCC patients with synchronous metastases. The PROBE trial (NCT04510597) is designed to complete the data gap about the impact of CN on outcomes in patients who started with an ICI-based combination treatment. After 9 to 12 weeks of systemic therapy, patients will be evaluated and randomized to undergo CN or to continue systemic therapy. The patients with disease progression will not be randomized. The primary endpoint of the trial is OS. The study hypothesizes that CN after initial ICI-based treatment will improve OS in patients with primary metastatic RCC [122]. The results are expected in 2033.

At this moment, we have to make treatment decisions on the basis of available retrospective data largely derived from institutional databases fraught with selection bias and unrecognized confounders and wait for the results from ongoing prospective trials. Discussion of patient cases in the multidisciplinary team remains crucial in the decisions regarding patient selection as well as the timing of the surgery. CN in appropriately selected patients could increase the efficacy of systemic therapy leading to improved OS. In patients with a surgically resectable primary tumor as well as resectable (or amenable to other focal therapies such as radiotherapy or ablation) metastases, surgical procedures leading to the complete removal of the tumor should always be discussed. The option of active surveillance can also be considered in selected cases [117,123]. When complete resection is obtained, subsequent adjuvant pembrolizumab should follow to increase the chance for a long-lasting remission. It should be kept in mind that resection of metastases may delay systemic therapy initiation and also spares the patients from adverse events complicating long-term systemic treatment. On the other hand, although a prolongation of OS is expected, CN in the setting of residual disease represents a palliative procedure. In patients with significant local symptoms and a surgically operable tumor, the benefit of CN is obvious. Moreover, considering the approach with deferred CN as reasonable, we should thoroughly evaluate the patient’s clinical characteristics and laboratory parameters along with the assessment of the response obtained with the upfront systemic treatment.

To achieve the optimal therapeutic goal, we should prioritize the selection of the optimal systemic regimen to be used in the context of CN. Since the data from clinical trials reported are very heterogenous in terms of ICI-based combinations used, it is possible that some regimens are superior to others in achieving a prolonged response to systemic therapy and CN, but no study so far has addressed this issue. For example, the CLEAR trial reported a complete response rate of 16% using a combination of pembrolizumab with lenvatinib [28], which is the highest number of CRs observed until now. There can be equivocal scenarios such as mixed response to ICIs which can complicate further decision making. The tumor is heterogenous and so is the tumor-immune microenvironment causing dynamic and often unpredictable treatment outcomes. ORR observed in trials with ICI-based combinations in a frontline setting has also been noted in patients with the primary tumors left in place [124]. Significant tumor shrinkage has been noted in more than 30% of cases in patients treated with avelumab and axitinib or nivolumab plus ipilimumab combinations [124,125]. Hence, starting with systemic therapy is a viable option, especially in patients with poor prognostic factors, or in cases in which immediate CN is not feasible. Once a response in the primary as well as the metastatic sites is obtained, CN that has so far been deferred can be offered. Not surprisingly, surgical complete remissions can be achieved including pathologic CRs [126,127,128]. Another important aspect is the discrepancy between the radiological outcome and the pathological findings in the resection specimen as described previously in the literature [129].

## 8. Conclusions

Facing the rapidly changing landscape of systemic therapy with continuous improvement of efficacy, the role of CN has been called into question. Irrespective of all the aspects that have to be considered in a multidisciplinary setting, patient selection is of paramount importance. Deferred CN has become the preferred and recommended approach with all its benefits in most cases where disease control was obtained. Undoubtedly, there are subgroups of patients who have long-lasting disease stabilization or even remissions as a result of ICI-based therapy in combination with CN. The issue of whether to consider CN in metastatic RCC patient clinical management remains not only a matter of controversy, but also, at the same time, the most important topic significantly affecting patient prognosis.

## Figures and Tables

**Table 1 cancers-15-03855-t001:** Selected trial results involving the role of CN in patients treated exclusively with ICIs or ICI-based combinations.

Author	Year	Number of Patients	Number CN	Number without CN	CN Systemic Therapy Sequence	Systemic Agents Used	OS in CN	OS without CN	nccRCC Included	HR	Ref.
Bakouny et al.	2022	437	234	203	upfront	IO, IO + TKI	54 mos	22 mos	Yes	0.61	[45]
Gross et al.	2023	367	232	135	upfront, deferred	IO	not reached (IQ3 33.1-NR)	14.9 mos (IQR 10.9–22.8)	Yes	0.33	[39]
Singla et al.	2020	391	221	170	upfront, deferred	IO	not reached	11.6 mos	No	0.23	[44]
Pignot et al.	2022	30	30	0	deferred	IO, IO + TKI	86.1% in 24 mos	NA	NA	NA	[47]
Rebuzzi et al.	2022	556	490	66	upfront	IO	35.9 mos	12.1 mos	Yes	0.44	[48]
Yoshino et al.	2022	41	21	13	deferred, upfront	IO	1-year deferred 100% vs. upfront 72.4%	58.2% 1-year survival	Yes	NA	[49]
Stellato et al.	2021	287	246	41	upfront	IO	20.9 mos	13 mos	Yes	0.64	[50]
Ghatalia et al.	2022	433	148	285	upfront, deferred	IO, TKI + IO	40.2 mos in upfront subgroup	15.2 mos	No	0.9 NS	[51]
Hahn et al.	2023	157	118	39	upfront, deferred	IO, TKI + IO	30.1 mos	13.3 mos	sarcomatoid and rhabdoid RCC only	0.79 NS	[52]

Ref., reference; CN, cytoreductive nephrectomy; HR, hazard ratio; IO, immunotherapy; mos, months; nccRCC, non-clear cell renal cell carcinoma; NS, not statistically significant; TKI, tyrosine kinase inhibitor.

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
