# Peer review of "The Evolving Landscape of Cytoreductive Nephrectomy in Metastatic Renal Cell Carcinoma"

_cancers, 2023, doi:10.3390/cancers15153855_

Round 1
Reviewer 1 Report
1. The beginning of section 3 is duplicated. Please review and correct.
2. Line 230. 'CN in this patient population should be per- 230 formed if feasible.' this comment is in contrast to the prior sentence. I agree that CN is appropriate in selected patients but please be consistent and clear in the message of the manuscript.
3. Line 337 is missing the citation.
4. Line 386. I think the authors should remove this statement of 'potentially curative' as no data suggests this to date. Theoretically it might be possible but again no data supports this statement.
5. Line 462. To be consistent please state the primary endpoint of the PROBE trial (OS).
6.
I recommend approval (after the minor corrections) as this is a very timely topic in kidney cancer.
Author Response
- The beginning of section 3 is duplicated. Please review and correct. Thank you for your comment. The text has been modified accordingly.
2. Line 230. 'CN in this patient population should be per- 230 formed if feasible.' this comment is in contrast to the prior sentence. I agree that CN is appropriate in selected patients but please be consistent and clear in the message of the manuscript.
Thank you for your comment. The sentence has been omitted.
3. Line 337 is missing the citation. It is the citation number 85 which is a line below.
4. Line 386. I think the authors should remove this statement of 'potentially curative' as no data suggests this to date. Theoretically it might be possible but again no data supports this statement. Thank you for your comment. The text has been modified accordingly.
5. Line 462. To be consistent please state the primary endpoint of the PROBE trial (OS).
The primary endpoint of the PROBE trial has been included.
I recommend approval (after the minor corrections) as this is a very timely topic in kidney cancer.
Thank you!

Reviewer 2 Report
In this narrative review article, Studentova et al. provide an overview of the current landscape of cytoreductive nephrectomy (CN) in the management of metastatic renal-cell carcinoma (mRCC).
The authors provide a comprehensive analysis of the role of CN aross the three different treatment eras of mRCC, describe its utility in non-clear cell and sarcomatoid histology, and explore aspects related to optimizing patient selection to decrease surgical morbidity and mortality and improve outcomes.
Overall, the authors have done a remarkable job summarizing the current literature on the role of CN in mRCC. I only have two minor suggestions to make:
1) Although the exact biologic mechanism through which CN may provide a survival benefit in mRCC patients is unclear, there are some suggested hypotheses [1]. I would advise the authors to briefly mention these potential mechanisms in the introduction section.
2) The paragraph referring to tumor size (line 180) is better suited for the "Surgical aspects of cytoreductive nephrectomy" section rather than the immunotherapy section.
References:
[1] Esagian SM, Ziogas IA, Kosmidis D, Hossain MD, Tannir NM, Msaouel P. Long-Term Survival Outcomes of Cytoreductive Nephrectomy Combined with Targeted Therapy for Metastatic Renal Cell Carcinoma: A Systematic Review and Individual Patient Data Meta-Analysis. Cancers (Basel). 2021 Feb 9;13(4):695. doi: 10.3390/cancers13040695. PMID: 33572149; PMCID: PMC7915816.
Moderate language editing is necessary to improve the clarity of the manuscript. Consider using shorter and simpler phrases.
Example:
"Due to the detection of small RCC lesions, because of widespread use of ultrasound and 50 computed tomography (CT), an increasing number of patients with RCC is being cured for early stages"
can be rephrased to:
"The widespread use of ultrasound and computed tomography has led to an increasing proportion of RCCs detected and treated at an early stage"
Consider applying similar changes to other parts of the manuscript where complex phrases may affect clarity.
Author Response
1) Although the exact biologic mechanism through which CN may provide a survival benefit in mRCC patients is unclear, there are some suggested hypotheses [1]. I would advise the authors to briefly mention these potential mechanisms in the introduction section. Thank you for your comment. Text has been modified accordingly and the mechanism of CN has been implemented in the section 2.
2) The paragraph referring to tumor size (line 180) is better suited for the "Surgical aspects of cytoreductive nephrectomy" section rather than the immunotherapy section.
Thank you for your comment. The text has been adapted accordingly and the paragraph has been moved to the surgical section.
Moderate language editing is necessary to improve the clarity of the manuscript. Consider using shorter and simpler phrases.
Example:
"Due to the detection of small RCC lesions, because of widespread use of ultrasound and 50 computed tomography (CT), an increasing number of patients with RCC is being cured for early stages"
can be rephrased to:
"The widespread use of ultrasound and computed tomography has led to an increasing proportion of RCCs detected and treated at an early stage"
Consider applying similar changes to other parts of the manuscript where complex phrases may affect clarity.
Thank you for your comment. We have tried to make the appropriate changes throughout the text.
